# Antimicrobial Activity of *Pinus wallachiana* Leaf Extracts against *Fusarium oxysporum* f. sp. *cubense* and Analysis of Its Fractions by HPLC

**DOI:** 10.3390/pathogens11030347

**Published:** 2022-03-12

**Authors:** Qurat Ul Ain, Shahzad Asad, Karam Ahad, Muhammad Naeem Safdar, Atif Jamal

**Affiliations:** 1PARC Institute of Advanced Studies in Agriculture, National Agricultural Research Centre, Islamabad 45500, Pakistan; 2Crop Diseases Research Institute, National Agricultural Research Centre, Islamabad 45500, Pakistan; asadtaimoor@yahoo.com; 3Institute of Plant and Environmental Protection, National Agricultural Research Centre, Islamabad 45500, Pakistan; karam_ahad@yahoo.com; 4Food Science Research Institute, National Agricultural Research Centre, Islamabad 45500, Pakistan; naeemsafdar03@yahoo.com

**Keywords:** *Fusarium oxysporum* f. sp. *cubense*, in vitro assay, fractions, high performance liquid chromatography, polyphenolic standards

## Abstract

Fusarium wilt has ruined banana production and poses a major threat to its industry because of highly virulent *Fusarium oxysporum* f. sp. *cubense* (Foc) race 4. The present study focused on the efficacy of *Pinus wallachiana* leaf extracts and its organic fractions against Foc in in vitro and greenhouse experiments. The presence of polyphenols in the fractions was also investigated using high performance liquid chromatography (HPLC). The in vitro tests carried out for the leaf extract of *P. wallachiana* showed its inhibitory effect on the mycelial growth and, based on this evidence, further characterization of fractions were done. Complete mycelial inhibition and the highest zone of inhibition against Foc was observed for the n-butanol fraction in vitro, while the n-hexane and dichloromethane fractions showed lower disease severity index (DSI) in greenhouse experiments. The fractions were further analysed by HPLC using nine polyphenolic standards, namely quercitin, myrecitin, kaempferol, rutin, gallic acid, trans-ferulic acid, coumeric acid, epicatechin and catechin. The highest content of polyphenols, based on standards used, was quantified in the n-butanol fraction followed by the ethyl acetate fraction of the leaf extract. This is the first report of antimicrobial activity of *Pinus wallachiana* extracts against Foc to the best of our knowledge.

## 1. Introduction

Banana (*Musa* spp.) is tremendously important for millions of cultivators and corporate growers, both for export and subsistence. The yield of commercial bananas across the world is staggeringly affected by Fusarium wilt (Panama disease) of bananas. It is a soil-borne disease whose causative agent is a hyphomycete i.e., *Fusarium oxysporum* f. sp. *cubense* [1,2,3]. Obliteration of Gros Michel by Foc race 1 led to its substitution with resistant Cavendish cultivars that are now susceptible to Foc race 4, specifically Foc TR4, which gained an emplacement from South East Asia to Africa and recently became entrenched in Latin America, thereby jeopardizing intercontinental banana production [4,5,6]. Management actions including crop rotation, flood fallowing, organic amendments, intercropping, molecular and biological control, etc. have been applied to combat this disease, but these measures provide short-term or little success under field conditions that advocate for continuous exploitation of belligerent methodologies that are antagonistic to the disease [6,7,8,9,10].

Various research investigations of plant crude extracts revealed their inhibitory activities against phytopathogens that account for the presence of antimicrobial secondary metabolites as their compositional constituents. Additionally, these secondary metabolites e.g., terpenoids, alkaloids, tannins, saponins, phenylpropanoids and flavanoids etc. are vital materials in the manufacture of sundry fungicides and pesticides [11,12,13,14,15]. Secondary metabolites signify the adaptive potential of plants against biotic and abiotic stresses [16]. Secondary metabolites’ structure, optimized through evolution, interferes with microbes molecular targets, hence acting as a mechanism for plant defense [17]. Phenolics are the profusely found secondary metabolites in plants [18]. Detection and identification of phenolics have now become an extensive research area because of the evidence that they have an indispensable role in the avoidance of the diseases that are linked to oxidative stress [19,20,21]. The plant phenolic compounds are studied as vital sources of novel antibiotics, insecticides, natural drugs and herbicides [22,23]. Continuous exploitation of botanicals from various plants and their different parts would be productive in discovering innovative, environmentally safe antimicrobials that can vanquish the complications of multi-drug resistance and bioaccumulation of pesticides.

Being used as folk medicines, gymnosperm botanicals have also been extensively studied for their anti-inflammatory and antimicrobial potential in recent decades. The presence of diverse chemical constituents in these extracts is thought to be responsible for microbial growth inhibition [24,25,26,27]. The *P. wallachiana* (commonly called Biar or Blue Pine) is a large cone-bearing evergreen tree belonging to the family Pinaceae of gymnosperms with a height up to 35–50 m and a diameter of 1–1.5 m, having down-curved branches with a straight trunk. Leaves are long (15–20 cm), slender, in fascicles of five, and flexible, the adaxial side having multiple bluish-white stomatal lines and the abaxial side green ones [28,29]. It is one of the principal conifers, mostly growing in the upper region of mountains associated with other gymnosperms, and is regarded as an important medicinal plant [30]. The majority of the research and pharmacognostic studies conducted on *P. wallachiana* have strongly supported its antioxidant efficacies [31,32,33,34] and the anticancerous potential of *P. wallachiana* needle extract [35]. The antibacterial activity of *P. wallachiana* essential oil against tested bacterial strains [36] and antifungal efficacy of its essential oil against *Fusarium verticillioides* [37], antimicrobial activity of its hydroalcoholic extracts against tested bacterial strains and fungi [38], and antibacterial activity against *Acinetobacter baumannii* [29] demonstrate its antimicrobial potential. Phytochemical studies have reported the antioxidant activity of *P. wallachiana* extracts that accounts for the presence of plentiful flavanoids and polyphenols in their phytochemical composition [32,39]. Phenolic compounds, i.e., chlorogenic acid, catechins, ferulic acid, and caffeic acid are well-known toxic compounds that are much faster concentrated in resistant varieties after their infection by the pathogen [40]. Cell wall phenolics, e.g., coumaric acid and trans ferulic acid, play a crucial role during plant growth by defending it against stresses including infections and wounding, etc. [41]. The antiviral potential of catechins and (-)-epicatechin gallate against the influenza virus had been noted. These polyphenols alter the membrane’s physical properties of the virus [42]. The antimicrobial potential of polyphenols e.g., catechin, gallic acid, ferulic acid, p-coumaric acid, quercitin, and rutin against *Xylella fastidiosa* had also been described earlier [43]. Similarly, the antifungal activities of polyphenolics e.g., phenol, catechin, quercetin, ο-coumaric acid, gallic acid, pyrogallic acid, ρ-coumaric acid, ρ-hydroxy benzoic acid, protocatechuic, salicylic acid, coumarin, and cinnamic acid had been noted [44]. Moreover, powerful antimicrobial activities by polyphenol compounds including kaempferol, gallic acid, quercetin and ellagic acid have been reported [45]. Extracts abundant in antioxidants, i.e., ascorbic acid, polyphenols and flavonoids, are a source of cell damage and the leaking of biomolecules from the impaired microbial membranes. The present study was designed to investigate the antifungal potential of *Pinus wallachiana* botanicals against Foc and evaluating its various fractions for the presence of some important polyphenols that might be beneficial for combating the Fusarium wilt problem.

## 2. Results

### 2.1. Fungicidal Analysis

The MIC and MFC of *P. wallachiana* extract against Foc were determined to be 20 mg/mL and 40 mg/mL respectively, while IC_50_ was calculated to be 6.09 mg/mL using regression equation (Table 1).

### 2.2. Effects on Biomass Production

Although *P. wallachiana* extract supplemented treatments (IC_50_, MIC, MFC) showed considerable reduction in Foc biomass compared to the control, maximum biomass reduction and 100% inhibition was found for MFC i.e., 40 mg/mL (Table 2).

### 2.3. Fractions of P. wallachiana

#### 2.3.1. The Percentage Yield of Fractions

Maximum yield was recorded for dichloromethane fraction (27.8%), followed by the n-butanol (25.12%), ethyl acetate (24.68%), and n-hexane (21.8%) fractions (Table 3).

#### 2.3.2. Antifungal Assays of Fractions

In the food poisoning assay, all fractions of *P. wallachiana* effectively inhibited mycelial growth of Foc compared to the solvent controls. The n-butanol fraction of *P. wallachiana* completely inhibited mycelial growth (i.e., 100%) followed by dichloromethane fraction (75.96%), n-hexane fraction (68.93), and ethyl acetate (57.26%) fraction (Table 3 and Appendix A). In well diffusion assay, the maximum zone of inhibitions was measured for n-butanol (24.4 mm) and dichloromethane fraction (23.8 mm), while n-hexane and ethyl acetate recorded 21 mm and 18.6 mm ZOI respectively (Table 4 and Appendix A).

#### 2.3.3. Greenhouse Experiment of Fractions

First severity scoring (based on a 1–5 scale) was performed after a month of first drenching. The highest disease severity index (DSI) value, calculated from the severity scores, was recorded for the n-butanol fraction (40 mg/mL) while the n-hexane fraction (20 mg/mL) along with the dichloromethane fraction (20 mg/mL) displayed the lowest DSI. Second drenching was applied after recording the first severity scoring and second severity scoring was performed after two months of second drenching. Maximum DSI i.e., 100% value was calculated by all solvent control treatments including fungicide (200 µg/mL) and n-butanol fraction (20 mg/mL). After second severity scoring, third drenching was applied. Third severity scoring was performed after four months of third drenching. The lowest DSI was noted for dichloromethane (20 mg/mL) and hexane (40 mg/mL) fractions with 60% values. Comparison of the DSI of different treatments, calculated at three different intervals, revealed that the progress of wilting was delayed in the case of dichloromethane (20 mg/mL) and hexane (40 mg/mL) fractions. Except for n-hexane, all the other fractions recorded maximum DSI in their higher concentration, i.e., 40 mg/mL (Table 5 and Appendix A).

### 2.4. HPLC of Fractions

The identification and quantification of polyphenolic compounds i.e., phenolic acids and flavonoids, were determined in the four fractions of *P. wallachiana* using HPLC analysis. Identification and quantification of phenolics (285 nm) and flavanoids (370 nm) was undertaken according to retention time (RT) and peak spectral characteristics against those of standards. Detection of polyphenolic compounds compared to standards and the overall polyphenolic content of *P. wallachiana* leaf extract varied in different fractions, as evident from the data (Table 6 and Appendix A). The HPLC chromatograms of polyphenolic standards and two fractions of *P. wallachiana* i.e., ethyl acetate and n-butanol showed that all the polyphenolic compounds were detected in the n-butanol and ethyl acetate fractions except for rutin. Likewise, only quercitin and ferulic acid were detected in the n-hexane fraction, while the dichloromethane fraction detected all polyphenolic compounds except rutin, myrecitin and catechin (Figure 1 and Appendix A).

Highest gallic acid (11.57 mg/g), catechin (33.44 mg/g), epicatechin (16.74 mg/g) and coumeric acid (4.33 mg/g) were detected in n-butanol fraction whereas highest ferulic acid (2.84 mg/g), myrecitin (2.15 mg/g), quercitin (7.9 mg/g) and kaempferol (7.81 mg/g) were quantified in ethyl acetate fraction. Maximum polyphenolic content, based on 9 polyphenol standards, were determined for n-butanol fraction of *P. wallachiana* (68.52 mg/g of extract) followed by ethyl acetate fraction (43.90 mg/g of extract) (Table 6).

## 3. Discussion

Due to the ethnopharmacological properties of plants, up to 50% of novel drugs are procured from natural sources [46,47]. Distinct plants and their different parts are administered in various modes for the treatment of infectious pathologies [48,49,50]. The active constituents of botanicals may take direct action on the pathogen or induce systemic resistance in the host plant that results in the decrement of disease development [51,52]. In the present investigation, the antimicrobial potential of *Pinus walliachina*; a gymnosperm, was explored against one of the most devastating pathogens, *Fusarium oxysporum* f. sp. *cubense*. Initial screening was carried out with *P. walliachina* leaf extracts for testing the antimicrobial potential. Results indicated that the extract effectively inhibited the growth of Foc, and based on these observations, further experiments were initiated which included the extraction of fractions using four solvents *viz*. hexane, dichloromethane, ethyl acetate and n-butanol and their potential to inhibit fungus in in vitro and in greenhouse. Both assay results verified the effectiveness of *P. walliachina* and, therefore, this study constitutes the first report of the antimicrobial activity of *P. walliachina* against Foc to the best of our knowledge. HPLC was also carried to further characterize all fractions, and nine standards were used for the said purpose.

The leaf extract completely inhibited the mycelial growth of Foc, and this observation was similar to the ones made previously where antifungal efficacy of extracts from distinct species associated with different families of gymnosperms was demonstrated [26] and also the efficacy of botanicals extracted from *P. walliachina* exhibited prominent antifungal, antibacterial and insecticidal activities [31,37,38,53]. The four fractions of *P. walliachina* leaf extract recorded significant percent inhibition and zone of inhibition against Foc in the poisoned food and well diffusion assays, respectively. These results are inconsistent with another study where fractions of *P. walliachina* crude leaf extract showed insecticidal (ethyl acetate) and antimicrobial (n-hexane) activities against *Rhyzopertha dominica* and *Microsporum cannis,* respectively [54]. The n-butanol (followed by the dichloromethane fraction) was found to be the most efficient treatment. However, the inhibitory potential of the four fractions was variable, which might be due to the different types of solvents used. It has been reported that the type of plant/plant part and type of extraction solvent are the reason for the variation of the phytochemical composition of various extracts [55].

In the greenhouse assay, the n-hexane fraction treatment with 40 mg/mL and dichloromethane fraction treatment with 20 mg/mL concentrations were found to be effective. The complete mycelial inhibition of Foc in the in vitro assay was observed in the n-butanol fraction, whereas in a greenhouse experiment the same fraction (40 mg/mL) recorded 100% DSI after one month of its very first drenching. It was noticed that polar fractions with their higher concentrations recorded comparatively higher DSI values, suggesting that with the high polarity of fraction its phytotoxicity to banana plantlets also increases. Polar fractions might have such phytochemicals that were not only detrimental to Foc but also had a phytotoxic effect on banana plantlets. A similar phytotoxicity phenomenon was described in an earlier study while working with different concentrations of chemical treatments (sterilant and fungicide) such as soil drenching. All chemicals with 50 μg/mL concentration developed severe phytotoxicity symptoms, while at lower concentrations none of the banana plantlets expressed phytotoxicity [56]. Similarly, in another study, significant phytotoxicity of various fractions of *P. wallachiana* leaves at 500 µg/mL were observed [54]. It can be concluded therefore that polar fractions should be used with comparatively lower concentrations i.e., less than 20 mg/mL, in order to decrease the DSI values.

The HPLC analysis of *P. wallachiana* fractions was done for the identification and quantification of polyphenolics using nine standards, and it confirmed the presence of most of the polyphenolic compounds in *P. wallachiana* fractions. All polyphenolic compounds except rutin were detected in the ethyl acetate and n-butanol fraction of *P. wallachiana* which corresponds with a past study that described that all pine extracts contain a high number of polyphenols [57,58,59]. The dichloromethane fraction detected all polyphenolic compounds except rutin, myrecitin and catechin, while the n-hexane fraction only detected ferulic acid and quercitin. Epicatechin, gallic acid, coumeric acid, and catechin were recorded the highest in n-butanol fraction, while kaempferol, ferulic acid, quercitin and myrecitin were detected highest in the ethyl acetate fraction. The highest polyphenolic content based on the nine polyphenolic standards was quantified for the *P. wallachiana* n-butanol fraction followed by ethyl acetate. An earlier study found quercetin as the most abundant flavonol in the n-butanol fraction (15.714%) of *P. wallachiana* methanol leaf extract using HPLC [60,61]. Similarly high amounts of polyphenolics, mainly taxifolin and catechins, were found to be the main reason for the antioxidant and biological activity of the *Pinus* species [62]. Moreover, phenolics and sulfur present in the plant extract contributed to the cell death of Foc TR4 by inducing oxidative bursts, mitochondrial impairment, and depolarization of the plasma membrane [63]. The production of phenolics in the resistant varieties of banana restricts pathogens to infected vessels due to lignifications of obstructions that result from the initial pathogen-induced occlusion reaction [64]. There is evidence that trans-ferulic acid and p-coumaric acid significantly inhibit the mycelial growth of Foc TR4 [65]. The presence of the polyphenolic compounds quantified in the fractions of *P. wallachiana* is the most probable reason for its mycelial inhibition activity against Foc.

## 4. Materials and Methods

### 4.1. Acquisition, Revival, and Confirmation of Fungal Culture

*Fusarium oxysporum* f. sp. *cubense* (Foc; TR4) was acquired from the Tissue culture department of National Agricultural Research Centre (NARC), Islamabad, the identity of which has been molecularly confirmed [66]. After the revival of Foc culture on potato dextrose agar (PDA), its morphology was examined, showing 3–5 septate, hyaline, sickle-shaped, macroconidia pointed at both ends and borne on single phialides, whereas microconidia were found to be mostly hyaline, kidney-shaped, aseptate produced on false heads.

### 4.2. Plant Sample and Extraction

A fresh leaf sample of *P. wallachiana* was collected from Ghora gali, Murree (altitude: 2291 m, coordinates 33°54′15″ N 73°23′25″ E) and after its disinfection with 5% Clorox, it was shade dried for 30 days and then was mechanically toiled. A powdered leaf sample was stored in labeled plastic jars for the in vitro assays that were performed in the fungal pathology laboratory of NARC. The leaf powder was mixed with ethanol using Erlenmeyer flasks, shaken at 60 rpm (revolution per minute) for 48 h and after its filtration, excess solvent was removed by the rotary evaporator [67], thereby dried extract was deposited in a glass vial [68].

### 4.3. Fungicidal Analysis

Twofold concentrations of the *P. wallachiana* leaf extract (1.25, 2.50, 5.0, 10, 20, and 40 mg/mL) were amended in autoclaved PDA media for the determination of minimum inhibitory concentration (MIC) and minimum fungicidal concentration (MFC) [69]. With the help of a plunger, 6 mm wells were made in the center of poisoned plates and Foc plugs were aseptically placed followed by the incubation (25 ± 2 °C) and recording of MIC and MFC after a week’-long interval. The half minimal inhibitory concentration (IC_50_) was also calculated using the regression equation [70].

### 4.4. Effect on Foc Biomass Production

The liquid culture was used to evaluate the effect of the extract on the production of Foc biomass [71]. Four treatments viz. control (no extract), IC50, MIC, MFC of the extract were separately dissolved in Potato Dextrose Broth (50 mL). Each flask aseptically received three to four plugs of Foc and placed on a rotary shaker (90 revolutions/min) and incubated (25 ± 2 °C) for a month. Mycelia-containing flasks were autoclaved and the media was filtered and mycelia were dried overnight (40 °C) after their washing with distilled water. Dry mycelia containing filter paper were then weighed and the percentage of growth inhibition was calculated by Equation (1) for each treatment as:P.I. = Dry weight of control − Dry weight of sample/Dry weight of control × 100(1)
where, P.I. = Percent inhibition

### 4.5. Fractionation

Liquid-liquid fractionation was performed for partitioning of *P. wallachiana* extract using a separating funnel [72]. The n-butanol, n-hexane, ethyl acetate, and dichloromethane were used as partitioning solvents. Fractionation was done in order of increasing polarity i.e., n-hexane > dichloromethane > ethyl acetate > n-butanol. The *P. wallachiana* extract was dissolved in water and sequential partitioning with n-hexane, dichloromethane, ethyl acetate, and n-butanol was done. Each fraction obtained was dried using a rotary evaporator and after calculation of its percentage yield using Equation (2), stored in labeled glass vials.
Yield = weight of dried fraction/initial weight of extract × 100(2)

### 4.6. Antifungal Assay of Fractions

#### 4.6.1. Food Poisoning Assay

Sterilized PDA plates poisoned with each fraction (10% conc.) and their 5% respective solvents that served as control were inoculated with Foc plugs (6 mm) and incubated at 25 ± 2 °C in five replicates [73]. When Foc mycelial growth completely covered all the control plates, radial mycelial growth was measured as the percentage inhibition of Foc using Equation (3).
P.I. = Radial mycelial growth of control − Radial mycelial growth of treatment/Radial mycelial growth of control × 100(3)
where, P.I. = Percent Inhibition

#### 4.6.2. Well Diffusion Assay

Spore suspension (10^6^) of Foc was spread on the entire surface of sterilized PDA as described earlier [74]. With the help of a cork borer, a hole with a diameter of 6 mm was punched aseptically in the center of 9 cm Petri plates (NEST, UK), and 100 µL from each fraction (10%) was introduced into the wells. Plates were incubated at 25 ± 2 °C in five replicates for all the treatments. The zone of inhibition (ZOI) started appearing after three days of incubation and was measured after one month.

### 4.7. Greenhouse Experiment

Dwarf Cavendish banana plantlets (six weeks old) were acquired from a tissue culture laboratory, NARC. A double pot system (15 cm × 15 cm × 12 cm) was used for banana plantation with a potting mixture of soil, sand, and peat moss in a 2:1:4 ratio. Millet grains colonized with Foc (50 g) were packed in the middle of potting mix in each double pot system [75] to serve as inoculum. Treatments were applied as soil drenching [56] after banana plantlet sowing. Two concentrations of fractions (20 mg/mL and 40 mg/mL) and propiconazole (100 µg/mL and 200 µg/mL) along with their respective controls were used as soil drench treatments.

Three drenchings were applied during the greenhouse experiment and assessment of visual symptoms was done after each drenching. Evaluation of disease severity based on visual symptoms was measured, [76] and the Disease Severity Index (DSI) for each treatment was calculated.

### 4.8. HPLC Analysis of Fractions

Nine polyphenolic standards were used in HPLC analysis for their detection and quantification in *P. wallachiana* fractions [77]. Fractions (1 mg/mL concentration) were filtered with the help of a Membrane filter (0.45 µm) and analyzed on a Perkin Elmer HPLC system equipped with a LC 295 UV/VIS detector, binary LC pump, and a reverse phase C18 column (4.6 mm × 250 mm, 5 µm). Solvent A (acetonitrile) and solvent B (distilled water/acetic acid, 99:1 *v*/*v*, pH 3.30 ± 0.1) were used in combination to serve as a mobile phase. Linear gradient mobile phase with a flow rate of 1 mL/min and 20 µL injection volume of the sample was employed with detector setting at 285 nm and 370 nm for phenolics and flavanoids, respectively. Gallic acid, epicatechin, catechin, trans-ferulic acid, and trans-p-coumaric acid were used as phenolic standards (Λ max: 285 nm). The conditions of gradient program used for phenolic acid separation were 20% A (5 min), 20% A (5 min), 80% A (10 min), 20% A (5 min). Flavonoids standards (Λ max: 370 nm) used were Rutin, Myrecitin, Quercitin and Kaempferol, and flavanoids were separated using the program: 20% A (5 min), 20% A (5 min), 80% A (7 min), 20% A (8 min). The analytes were identified by comparing the Rt (retention time), and spike samples with polyphenolic standards and subsequent quantification of phenolic compounds was determined.

## 5. Conclusions

This study exclusively evaluated *P. wallachiana* and its fractions efficacy against Foc and noted significant antifungal activity using in vitro and greenhouse assays, suggesting their potential role in the management of the vascular wilt of bananas. It was established that polyphenolic compounds have potent efficacy against phytopathogens, as we know that phenolic compounds are active in plant defense response. HPLC analysis of *P. wallachiana* fractions revealed the presence of most of the compounds (based on nine polyphenolic standards), with their maximum quantification in n-butanol and ethyl acetate fractions. The existence of such important polyphenols with known antimicrobial properties accounts for the antifungal efficacy of the *P. wallachiana* fractions against Foc and this scientific finding has not been reported prior to this study. The research study strongly recommends that *P. wallachiana* and its fractions be exploited further for the presence of valuable compounds that can make a breakthrough for the control of Panama wilt disease soon.

## Figures and Tables

**Figure 1 pathogens-11-00347-f001:**
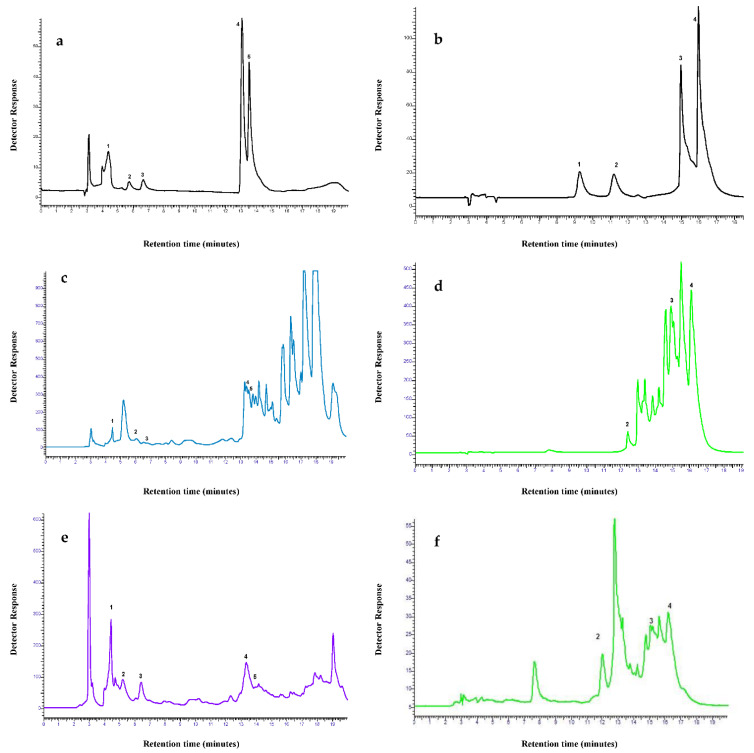
(**a**) Typical chromatogram of polyphenol standards (100 ppm) at 285 nm. 1 = Gallic acid, 2 = Catechin, 3 = Epicatechin, 4 = Coumaric acid, 5 = trans-Ferulic acid. (**b**) Typical chromatogram of flavanoids (100 ppm) at 370 nm. 1 = Rutin, 2 = Myrecitin, 3 = Quercetin, 4 = Kaempferol (**c**) Chromatogram obtained for ethyl acetate fraction at 285 nm. 1 = Gallic acid, 2 = Catechin, 3 = Epicatechin, 4 = Coumaric acid, 5 = trans-Ferulic acid, (**d**) Chromatogram obtained for ethyl acetate fraction of at 370 nm. 2 = Myrecitin, 3 = Quercitin, 4 = Kaempferol, (**e**) Chromatogram obtained for n-butanol fraction at 285 nm. 1 = Gallic acid, 2 = Catechin, 3 = Epicatechin, 4 = Coumaric acid, 5 = trans-Ferulic acid, (**f**) Chromatogram obtained for n-butanol fraction at 370 nm. 2 = Myrecitin, 3 = Quercitin, 4 = Kaempferol.

**Table 1 pathogens-11-00347-t001:** Determination of Half minimal inhibitory concentration (IC_50_) of *P. wallachiana* against Foc.

*P. wallachiana* Leaf Extract (Concentration in mg/mL)	Percent Inhibition	IC_50_	R^2^	Regression Equation
1.25	25 ± 1.20	6.09	0.9435	y = 3.7999x + 26.853
2.5	32.2 ± 0.7			
5	54.5 ± 0.5			
10	71.6 ± 0.3			
20	98.3 ± 0.4			

**Table 2 pathogens-11-00347-t002:** Effect of *P. wallachiana* extract on the biomass production of Foc.

Treatments	Biomass Production	
	Dry Weight (mg)	Percent Inhibition
Control (0)	158	0.00
IC_50_ (6.09)	58.7	62.9
MIC (20)	2.4	98. 5
MFC (40)	0	100

**Table 3 pathogens-11-00347-t003:** Percentage yield of four fractions of *P. wallachiana* prepared through liquid-liquid fractionation.

Fractions	Percentage Yield (%)
n-Hexane fraction	21.8
Dichloromethane fraction	27.8
Ethyl acetate fraction	24.68
n-Butanol fraction	25.12

**Table 4 pathogens-11-00347-t004:** Percent inhibition and zone of inhibition values recorded for *P. wallachiana* fractions against Foc using in vitro assays.

Treatments	Percent Inhibition	Zone of Inhibition (ZOI)
n-Hexane control	0.00 ± 0.00 ^E^	0.00 ± 0.00 ^D^
n-Hexane fraction	68.93 ± 0.47 ^C^	21.0 ± 0.92 ^B^
Dichloromethane control	0.00 ± 0.00 ^E^	0.00 ± 0.00 ^D^
Dichloromethane fraction	75.96 ±0.30 ^B^	23.80 ± 1.12 ^A^
Ethyl acetate control	0.00 ± 0.00 ^E^	0.00 ± 0.00 ^D^
Ethyl acetate fraction	57.26 ± 0.39 ^D^	18.60 ± 0.51 ^C^
n-butanol control	0.00 ± 0.00 ^E^	0.00 ± 0.00 ^D^
n-butanol fraction	100 ± 0.00 ^A^	24.40 ± 0.43 ^A^

Data presented as mean value of five replicates ± represents standard error. Significant differences among treatments were indicated by different superscript letters within individual columns.

**Table 5 pathogens-11-00347-t005:** Comparison of the three severity scorings and their respective disease severity indices calculated for banana plants drenched with fraction treatments in three different intervals during a greenhouse experiment.

Treatments	First Drenching		Second Drenching		Third Drenching	
	1st Severity Score	DSI	2nd Severity Score	DSI	3rd Severity Score	DSI
Simple Control	4.286 ± 0.29 ^BC^	85.71	5.000 ± 0.00 ^A^	100	5.000 ± 0.00 ^A^	100
Fungicide (100 µg/mL) Conc. 1	3.429 ± 0.20 ^DE^	68.57	4.000 ± 0.31 ^BC^	80	3.857 ± 0.34 ^B^	77.14
Fungicide (200 µg/mL) Conc. 2	4.286 ± 0.29 ^BC^	85.7	5.000 ± 0.00 ^A^	100	5.000 ± 0.00 ^A^	100
Hexane Control	3.714 ± 0.29 ^CDE^	74.28	5.000 ± 0.00 ^A^	100	5.000 ± 0.00 ^A^	100
Hexane (20 mg/mL) Conc. 1	2.286 ± 0.18 ^G^	45.71	2.571 ± 0.37 ^F^	51.43	3.571 ± 0.37 ^BC^	71.43
Hexane (40 mg/mL) Conc. 2	2.571 ± 0.20 ^G^	51.43	2.571 ± 0.30 ^F^	51.43	3.000 ± 0.22 ^C^	60
Dichloromethane Control	4.286 ± 0.36 ^BC^	85.71	5.000 ± 0.00 ^A^	100	5.000 ± 0.00 ^A^	100
Dichloromethane (20 mg/mL) Conc. 1	2.286 ± 0.29 ^G^	42.85	2.857 ± 0.26 ^EF^	57.14	3.000 ± 0.38 ^C^	60
Dichloromethane (40 mg/mL) Conc. 2	3.571 ± 0.20 ^DE^	71.43	3.714 ± 0.29 ^CD^	74.28	3.571 ± 0.53 ^BC^	71.43
Ethyl acetate Control	4.000 ± 0.22 ^BCD^	80	5.000 ± 0.00 ^A^	100	5.000 ± 0.00 ^A^	100
Ethyl acetate (20 mg/mL) Conc. 1	2.714 ± 0.18 ^FG^	54.28	4.571 ± 0.30 ^AB^	60	4.286 ± 0.29 ^AB^	71.43
Ethyl acetate (40 mg/mL) Conc. 2	3.286 ± 0.29 ^EF^	65.71	5.000 ± 0.00 ^A^	65.71	5.000 ± 0.00 ^A^	74.28
n-butanol Control	4.571 ± 0.20 ^AB^	91.43	5.000 ± 0.00 ^A^	100	5.000 ± 0.00 ^A^	100
n-butanol (20 mg/mL) Conc. 1	4.429 ± 0.20 ^AB^	88.57	3.000 ± 0.31 ^EF^	91.43	3.571 ± 0.37 ^BC^	85.71
n-butanol (40 mg/mL) Conc. 2	5.000 ± 0.00 ^A^	100	3.286 ± 0.29 ^DE^	100	3.714 ± 0.36 ^BC^	100

Same superscript letters within an individual severity scores column do no differ statistically and a common letter sharing between the treatments indicates non-significant difference. The disease severity index (DSI) was calculated by using formula. Seven replicates for each treatment.

**Table 6 pathogens-11-00347-t006:** Phenolic compound profile of the four fractions of *P. wallachiana* quantified through HPLC analysis.

Phenolic Compounds (mg/g of Extract)	n-Hexane Fraction	Dichloromethane Fraction	Ethyl Acetate Fraction	n-Butanol Fraction
Gallic acid	N.D.	0.10 ± 0.0033	3.57 ± 0.016	11.57 ± 0.0089
Catechin	N.D.	N.D.	13.46 ± 0.007	33.44 ± 0.0087
Epicatechin	N.D.	1.19 ± 0.0053	3.23 ± 0.0090	16.74 ± 0.0074
Coumeric acid	N.D.	0.61 ± 0.0043	2.94 ± 0.0068	4.33 ± 0.0034
Trans-Ferulic acid	0.13 ± 0.0004	0.61 ± 0.0037	2.84 ± 0.0039	0.52 ± 0.0018
Rutin	N.D.	N.D.	N.D.	N.D.
Myrecitin	N.D.	N.D.	2.15 ± 0.0044	0.74 ± 0.0064
Quercitin	0.04 ± 0.00001	0.06 ± 0.0005	7.9 ± 0.0056	0.52 ± 0.0041
Kaempferol	N.D.	0.09 ± 0.0034	7.81 ± 0.011	0.66 ± 0.0058
Total Polyphenolic Content	0.17 mg/g	2.66 mg/g	43.90 mg/g	68.52 mg/g

Values are mean of three replications. N.D. = Not detected.

## Data Availability

Not applicable.

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
