# Peer review of "Antimicrobial Activity of Pinus wallachiana Leaf Extracts against Fusarium oxysporum f. sp. cubense and Analysis of Its Fractions by HPLC"

_pathogens, 2022, doi:10.3390/pathogens11030347_

Round 1
Reviewer 1 Report
The manuscript entitled "Antimicrobial activity of Pinus wallachiana against Fusarium oxysporum f. sp. cubense and analysis of its fractions by HPLC" had the purpose to evaluate the efficacy of Pinus wallachiana and its organic fractions against Fusarium oxysporumn in vitro. Th manuscript is overall well written and brings some novelty to the natural compounds field.
The main issue regarding the manuscript is that the fungicidal activity was evaluated using P. wallachiana leaf extract and not the effects of the fraction compounds and, in doing so, it is not possible to realize which is the effective compound.
Author Response
Reviewer 1
Comment: The manuscript entitled "Antimicrobial activity of Pinus wallachiana against Fusarium oxysporum f. sp. cubense and analysis of its fractions by HPLC" had the purpose to evaluate the efficacy of Pinus wallachiana and its organic fractions against Fusarium oxysporum in vitro. The manuscript is overall well written and brings some novelty to the natural compounds field.
Answer: Thank you.
Comment: The main issue regarding the manuscript is that the fungicidal activity was evaluated using P. wallachiana leaf extract and not the effects of the fraction compounds and, in doing so, it is not possible to realize which is the effective compound.
Answer: We agree that this study doesn’t confirm that which compound(s) is responsible for fungal inhibition. We would like to submit that the main aim of our study was to investigate the effectiveness of P. wallachiana leaf extract against Foc and accordingly we planned this study with different experiments. We would ideally have done that but both financial resources and time were the limiting factors. This study is definitely in our future plans.
Reviewer 2 Report
Comments:
The title of the manuscript is "Antimicrobial activity of Pinus wallachiana against Fusarium oxysporum f. sp. cubense and analysis of its fractions by HPLC." However, technically and according to the results presented, it is not correct to affirm that the Pinus wallachiana plant is responsible per se for the evidenced antifungal activity. The experimental protocol used extracts obtained from the plant by being treated with different solvents, with n-butanol being the solvent that provided the most active extract under in vitro and greenhouse conditions. An organic extract corresponds to an organic compounds mixture, and their composition depends on many factors such as the used plant part, nutrients, maturation time, temperature, etc. Moreover, there is no evidence of experiments conducted in which Pinus wallachiana plants were cultivated in the same biological environment as the Cavendish banana plantlets, performing a "cultural" or "biological" control study that assures that the plant is responsible for the antifungal activity. Therefore, the title and part of the abstract should be modified, indicating that only the Pinus wallachiana extracts and not the plant itself are responsible for the antifungal activity. On the other hand, given the scope of the results, although using patterns to identify secondary metabolites in extracts is an appropriate practice, I consider that the HPLC study should be complemented by using a coupled technique such as HRMS that allows a more significant contribution and with this, subsequent studies evaluating the metabolites and their possible antifungal activity.
Author Response
Reviewer 2
Comment: The title of the manuscript is "Antimicrobial activity of Pinus wallachiana against Fusarium oxysporum f. sp. cubense and analysis of its fractions by HPLC." However, technically and according to the results presented, it is not correct to affirm that the Pinus wallachiana plant is responsible per se for the evidenced antifungal activity. The experimental protocol used extracts obtained from the plant by being treated with different solvents, with n-butanol being the solvent that provided the most active extract under in vitro and greenhouse conditions. An organic extract corresponds to an organic compounds mixture, and their composition depends on many factors such as the used plant part, nutrients, maturation time, temperature, etc. Moreover, there is no evidence of experiments conducted in which Pinus wallachiana plants were cultivated in the same biological environment as the Cavendish banana plantlets, performing a "cultural" or "biological" control study that assures that the plant is responsible for the antifungal activity. Therefore, the title and part of the abstract should be modified, indicating that only the Pinus wallachiana extracts and not the plant itself are responsible for the antifungal activity.
Answer: Thanks for reviewing it critically. We agree with your observations and therefore have modified the title and abstract accordingly. Briefly just want to elaborate two points for your perusal. The main aims of this study was to determine whether Pinus wallachiana has antifungal activity and for those reason fractions using different solvents in increasing polarity were used. Initially we have done screening of leaf extract and its MIC, MFC & IC50 were determined. Further its fractions were prepared using organic solvents in increasing polarity so as to extract maximum amount of possible compounds having variable polarity affinities for the used solvents. Comparable to the main leaf extract of Pinus wallachiana all its fractions i.e. n-hexane, dichloromethane, ethyl acetate and n-butanol significantly inhibited the growth of Foc. It is known fact that phenolic compounds effectively control plant pathogens so keeping this hypothesis in mind these fractions were evaluated by using renowned phenolic standards e.g. gallic acid, ferulic acid, coumeric acid, quercitin, kaempferol, myrecitin, catechin, epicatechin and rutin using HPLC. Presence of most of these phenolic compounds confirmed our hypothesis. We didn’t grow Piunus wallachiana plant rather we used fractions of leaf extract as drenching treatment for Cavendish banana plantlets inoculated with Foc. These two plants don’t share same ecological niche that’s why extracts were used.
Comment: On the other hand, given the scope of the results, although using patterns to identify secondary metabolites in extracts is an appropriate practice, I consider that the HPLC study should be complemented by using a coupled technique such as HRMS that allows a more significant contribution and with this, subsequent studies evaluating the metabolites and their possible antifungal activity.
Answer: It’s a good suggestion but unfortunately we don’t have HRMS in both, our institute or any other institute / university nearby and due to limited funds, we were unable to do analyses from abroad.
Round 2
Reviewer 1 Report
I believe the authors have showed the importance of making new studies up front.
Reviewer 2 Report
Thanks for your answer. I suggest accepting the corrected version of the manuscript.